# Cost effectiveness of temporary isolation rooms in acute care settings in Singapore

**Nicholas Graves**[1]*, **Yiying Cai**[1], **Brett Mitchell**[2], **Dale Fisher**[3,4], **Martin Kiernan**[2,5]

**1** Health Services & Systems Research, Duke-NUS Medical School, Singapore, Singapore, **2** School of Nursing and Midwifery, University of Newcastle, Ourimbah, NSW, Australia, **3** Yong Loo Lin School of Medicine, National University of Singapore, Singapore, Singapore, **4** Department of Medicine, National University Hospital, Singapore, Singapore, **5** Gama Healthcare Ltd, Hemel Hempstead, United Kingdom

* n.graves@duke-nus.edu.sg

## Abstract

### Objectives

To estimate the change to health service costs and health benefits from a decision to adopt temporary isolation rooms that are effective at isolating the patient within a general ward environment. We assess the cost-effectiveness of a decision to adopt an existing temporary isolation room in a Singapore setting.

### Method

We performed a model-based cost-effectiveness analysis to evaluate the impact of a decision to adopt temporary isolation rooms for infection prevention. We estimated changes to the costs from implementation, the number of cases of healthcare associated infection, acute care bed days used, they money value of bed days, the number of deaths, and the expected change to life years. We report the probability that adoption was cost-effective by the cost by life year gained, against a relevant threshold. Uncertainty is addressed with probabilistic sensitivity analysis and the findings are tested with plausible scenarios for the effectiveness of the intervention.

### Results

We predict 478 fewer cases of HAI per 100,000 occupied bed days from a decision to adopt temporary isolation rooms. This will result in cost savings of $SGD329,432 and there are 1,754 life years gained. When the effectiveness of the intervention is set at 1% of cases of HAI prevented the incremental cost per life year saved is $16,519; below the threshold chosen for cost-effectiveness in Singapore.

### Conclusions

We provide some evidence that adoption of a temporary isolation room is cost-effective for Singapore acute care hospitals. It is plausible that adoption is a positive decision for other countries in the region who may demonstrate fewer resources for infection prevention and control.

**Data Availability Statement:** All data used were publicly available from public sources or published papers except for the data collected for the HAI PPS surveillance study that were reanalysed for this publication. These data are not available due to

potentially identifying or sensitive patient information. This ethics review boards of National Health Group Singapore and Singapore Health Services imposes this restriction. Kyaw Zaw Linn, Surveillance Coordinator at National Center for Infectious Diseases can field queries; Zaw_Linn_KYAW@ncid.sg. The public sources or published papers we used are: WHO. Report on the Burden of Endemic Health Care-Associated Infection Worldwide. 2011. Available here https://apps.who.int/iris/bitstream/handle/10665/80135/9789241501507_eng.pdf; Ministry of Health Singapore. The national infection prevention and control guidelines for acute healthcare facilities. 2017. Available here https://www.moh.gov.sg/resources-statistics/guidelines/infection-control-guidelines/national-infection-prevention-and-control-guidelines-for-acute-healthcare-facilities; Roosa Tikkanen RO, Elias Mossialos, Ana Djordjevic, George A. Wharton,. International Health Care System Profiles, Singapore. Available here https://www.commonwealthfund.org/international-health-policy-center/countries/singapore2020; Salma Khalik Senior Health Correspondent. Health Ministry data shows patients are now staying longer in hospital. Available here https://www.straitstimes.com/singapore/health/health-ministry-data-shows-patients-are-now-staying-longer-in-hospital. Straits Times. 2014.

**Funding:** No grant funding was used for this project. NG was paid consulting fees by GAMA Healthcare to develop a model and prepare a first draft of the manuscript. Martin Kiernan supported many aspects of the study and write up, but he did this without bias and focused on the best interpretation of the data. No other individual from GAMA healthcare influenced the study methods, findings or interpretation.

**Competing interests:** Martin Kiernan is clinical director for GAMA Healthcare. This does not alter our adherence to PLOS ONE policies on sharing data and materials.

# Introduction

Healthcare associated infections (HAIs) caused by multidrug resistant organisms (MDROs) are a major concern in hospitals globally [1]. These organisms include methicillin resistant *Staphylococcus aureus* (MRSA), multi-drug resistant non-fermenters, carbapenem-resistant Enterobacterales (CRE), vancomycin resistant *Enterococci* and *Candida auris*. MRSA is a major healthcare-associated pathogen that is endemic in many healthcare settings and is associated with worse health outcomes and economic costs [2, 3]. Emerging threats such as carbapenemase-producing carbapenem-resistant Enterobacterales (CP-CRE) and MCR-1-producing colistin-resistant Enterobacterales have the potential for rapid spread making it critically important for aggressive infection prevention and control measures [3, 4].

Well planned infection prevention and control strategies are critical in preventing MDRO associated HAIs. Singapore is a major travel hub with many people arriving each year to access health services. Decision makers have responded to the threat from MDRO transmission in its healthcare institutions using an extensive range of infection prevention strategies [5]. Universal active surveillance for MRSA and targeted screening for other pathogens play an important role, while detailed manual and non-touch environmental cleaning strategies minimise the risk of spread from surfaces and equipment. Limiting infection transmission through the isolation of patients is another important strategy. There are five specific transmission pathways that could be interrupted by the effective isolation of individuals: patient to healthcare worker (HCW); patient to environment; HCW to patient; environment to patient; and, environment to HCW [6].

Even though the rationale for isolation of patients colonised and or infected with MDROs is strong it appears impractical to provide permanent single-room isolation facilities for all MDRO colonised and/or infected patients in Singapore acute care hospitals. The majority of acute beds in Singapore public hospitals are in 'Type B' or 'Type C' wards that comprise 4, 6 or 8 beds. Only ICUs, 'Type A' wards and specialised isolation rooms are single room with adjoining bathroom and toilet. Temporary isolation spaces were however established for treating COVID-19 patients when the pandemic unfolded in Singapore [7].

We investigate a possible role for temporary 'pop-up' isolation rooms that are effective at isolating the patient within a general ward environment. For this paper we consider 'Rediroom' a mobile cart that unpacks into an air-filtered isolation room that offers the users hands-free entry [8]. Given that resources for infection control are finite [9] there is a need to identify whether a decision to add this intervention would be cost-effective [10] when compared to current infection prevention efforts.

The study question is by how much are 'health services costs' and 'patient health benefits' are expected to change from a decision to implement a temporary isolation room into acute care hospitals in Singapore. This will be cosidered in a framework for cost-effectivceness analysis [11]. The finding will be useful for those managing hospitals in Singapore with endemic MDROs, many with increasing incidence, and inadequate isolation capacity [12].

# Methods

## Target population, setting and outcome measures

The target population for this study are adult admissions to acute care hospitals in Singapore who face risk of health care acquired infection. The Singapore health system has 2.4 acute beds per 1,000 population in nine government supported hospitals, eight for-profit hospitals and one not-for-profit hospital [13]. Block funding by the government is accompanied by some out of pocket charges to patients, but when individuals are unable to pay there is a government

financial safety net. Comprehensive specialist acute care services are available. We model outcomes in adult patients for healthcare associated sepsis, pneumonia, surgical site infection, central line associated blood stream infection (CLABSI), intra-abdominal infection and other types of HAI. Recent and high quality data are available for these events from the first Singapore national point prevalence survey [14].

The outcomes evaluated from a decision to adopt temporary isolation rooms are the changes to: number of patients with HAI; bed days used for HAI; monetary value of bed days used; number of deaths; and, number of discounted life years. These outcomes inform estimates of the change to 'total health service costs' and 'life years' from a decision to implement a temporary isolation room in the acute setting. Change to costs are divided by change to life years to show an incremental cost-effectiveness ratio [11]. All costs are for the financial year ending in March 2021.

## Perspective and comparators

The cost perspective is the health service. We compare the adoption of a temporary isolation room to the existing arrangements for infection prevention. The National Infection Prevention Committee, a partnership between Singapore's hospitals and the Ministry of Health sets national policies. They include the use of bedside alcohol-based hand rub, active surveillance for MRSA, vancomycin-resistant *Enterococci* and carbapenemase-producing carbapenem-resistant *Enterobacterales*, bundles for device care and surgical site infection, performance indicators, environmental cleaning protocols and non-touch technology. The time horizon for the analysis is 12 months so no discounting rate applies to costs, but health benefits measured in life years attract discounts of 3% per year [15]. Because the durations of HAI are relatively short the use of preference utility weights to show quality adjusted life year (QALYs) is unnecessary.

## Measurement of effectiveness

There are no data to describe the real-world effectiveness of temporary isolation rooms and so scenarios are tested. We assume on average 30% of cases of healthcare associated infection will be avoided, which is consistent with previous studies [16, 17]. We also analyse effectiveness by reducing the estimate in the model until the decision to adopt is not supported against the criterion of cost effectiveness. We seek the minimum effectiveness at which adoption is supported against the criterion of cost-effectiveness.

## Health outcomes and costs

Changes to health outcomes are characterised by the reduction in risk of mortality from avoiding a case of HAI. The data for attributable mortality for a case of HAI are from the first national point prevalence survey [14], Table 1.

A one-proportion z-test enabled an estimate and 95% confidence interval of the probability of death for each type of HAI, see S1 Appendix. These are not adjusted for other factors that might affect risk of death. The costs of a bed day in the public system were taken from Singapore Ministry of Health [18], Table 2. The costs of adoption comprise a monthly capital cost plus a single-use canopy cost per patient, with the estimates used shown in Table 2. All costs are relevant for 2018.

## Other parameters

Age and gender distribution of the patients and the risks of HAI are taken from the recent prevalence survey [14], and the excess length of stays arising from a case of HAI are taken

**Table 1. Hospital mortality outcomes for those with and without HAI.**

| Patient group | Died | Survived |
|---|---|---|
| All HAI (n = 469) | 134 | 335 |
| Sepsis (n = 142) | 37 | 105 |
| Pneumonia (n = 105) | 45 | 60 |
| Surgical (n = 115) | 22 | 93 |
| CLABSI (n = 40) | 10 | 30 |
| Intra-abdominal (n = 28) | 18 | 10 |
| Others (n = 39) | 29 | 10 |
| No HAI (n = 3,959) | 550 | 3409 |

CLABSI = central line associated bloodstream infection

**Table 2. Input parameters for the cost-effectiveness model.**

| Parameter | Estimate (SD) | Prior Distribution | Source |
|---|---|---|---|
| Cases of HAIs per 10,000 admissions | 1,598 (84) | Normal (1598, 84) | [14] |
| Average length of stay of all patients | 6.4 (1.6) | Gamma (16.00, 0.40) | [19] |
| *Probability of HAI type* | | | |
| Sepsis | 0.30 | Beta (135.58, 312.33) | [14] |
| Pneumonia | 0.22 | Beta (90.13, 312.44) | |
| Surgical site infection | 0.25 | Beta (124.60, 383.54) | |
| CLABSI | 0.09 | Beta (36.43, 390.73) | |
| Intraabdominal | 0.06 | Beta (24.90, 392.25) | |
| Others | 0.08 | Beta (35.47, 391.06) | |
| *Excess LOS of each HAI* | | | |
| Sepsis | 0.89 (0.40) | Gamma (4.85, 0.18) | [20] |
| Pneumonia | 3.14 (0.56) | Gamma (31.79, 0.10) | |
| Surgical site infection | 3.62 (0.64) | Gamma (31.94, 0.11) | |
| CLABSI | 2.99 (1.13) | Gamma (7.04, 0.42) | |
| Intraabdominal | 1.58 (1.01) | Gamma (42.48, 0.45) | |
| Others | 1.92 (0.93) | Gamma (4.28, 0.45) | |
| *Probability of death of each HAI* | | | |
| Sepsis | 0.26 | Beta (34.07, 96.70) | [14] |
| Pneumonia | 0.43 | Beta (41.90, 55.87) | |
| Surgical site infection | 0.19 | Beta (19.77, 83.56) | |
| CLABSI | 0.25 | Beta (8.76, 26.28) | |
| Intraabdominal | 0.36 | Beta (9.06, 16.30) | |
| Others | 0.26 | Beta (8.78, 25.46) | |
| Probability of death in patients without HAI | 0.14 | Beta (537.18, 3329.53) | |
| Cost per bed-day (in SGD) | 823 (277) | Gamma (8.78, 93.77) | [18] |
| Cost of canopy per admission (in SGD) | 975 | Fixed | [21] |
| Capital cost of cart per month (in SGD) | 1145 | Fixed | |
| Mean age of patients | 67.6 | Fixed | [14] |
| Male % | 51.9 | Fixed | [14] |
| *Life expectancy years* | | | |
| Male | 81.5 | Fixed | [22] |
| Female | 86.1 | Fixed | |
| % admissions isolated | 4 to 10 | Uniform | #, [23] |
| % effectiveness | 0.30 (0.05) | Beta (24.9, 58.1) | Assumption |

from a separate published analysis [20]. Life expectancy is taken from the Singapore census [22]. The proportion of admissions that could be isolated if the technology were adopted are the MRSA cases admitted into hospital not routinely isolated. Current infection prevention practices are to prioritise CP-CRE, VRE, C difficile, rotavirus, tuberculosis and other outbreak prone or high impact diseases and these patients once identified are always isolated.

### Dealing with uncertainty, threshold for cost-effectiveness and model evaluation

Uncertain parameters are characterised by prior statistical distributions and some values are fixed. All parameters are subject to 10,000 random samples to produce output distributions for the model outcomes. The threshold for cost-effectiveness was the mean GDP per capita, which is USD $59,798 or approximately SGD $80,000 [17]. This approach assumes one year of perfect quality life is worth the per capita gross domestic product [18]. We report the 'probability that an adoption decision is cost-effective' [24] and values for this statistic that exceed 50% suggest adoption is a better decision than remaining with current arrangements, yet values close to 50% imply large uncertainty in the decision and more information may be required prior to an implementation decision being made [25].

### Scenario analyses

The attributable mortality is unadjusted for other factors that can affect mortality. To investigate the robustness of the model conclusions to this parameter we halve the estimates of attributable mortality, reducing the health benefits from a decision to adopt, and re-examine the findings. A CHEERS checklist has been completed and included as an appendix.

## Results

The expected changes to the outcomes from a decision to adopt a temporary isolation room at an assumed effectiveness of 30% reduction in cases are shown in Table 3. On average there will be 478 fewer cases of HAI per 100,000 occupied bed days from a decision to adopt a temporary isolation room. This will release 1,627 bed-days for other uses, and these are valued at SGD $1.33M in savings. One hundred and thirty-six lives will be saved and 1,754 life years gained.

The joint distribution of expected change 'total costs' and 'life years' gained is shown in Fig 1. The mean change to total costs is -$SGD329,432, indicating overall that the cost savings from fewer cases HAI exceed the implementation costs. For this cost saving there are 1,754 life years gained. There is a 67% probability that adoption will be cost saving and 100% probability it will be cost-effective against the threshold value of $SGD80,000 per life year gained.

**Table 3. Changes to outcomes from a decision to adopt a temporary isolation rooms, per 100,000 occupied bed days.**

| Mean (sd) | Cases HAI | bed days | Money value of bed days (SGD) | number of deaths | life years |
|---|---|---|---|---|---|
| ALL HAI | 478 (83) | 1627 (338) | $1,325,570 ($548,774) | 136 (26) | 1754 (333) |
| Sepsis | 145 (27) | 127 (64) | $104,330 ($67,669) | 38 (9) | 483 (114) |
| Pneumonia | 107 (22) | 342 (91) | $280,396 ($126,657) | 46 (11) | 591 (140) |
| Surgical | 117 (22) | 414 (108) | $336,915 ($149,410) | 22 (6) | 289 (79) |
| CLABSI | 41 (10) | 121 (55) | $98,173 ($56,511) | 10 (4) | 132 (49) |
| Intra-abdominal | 28 (7) | 547 (168) | $442,796 ($212,695) | 10 (4) | 130 (49) |
| Others | 40 (21) | 76 (57) | $62,958 ($55,843) | 10 (6) | 130 (79) |

CLABSI = central line associated bloodstream infection

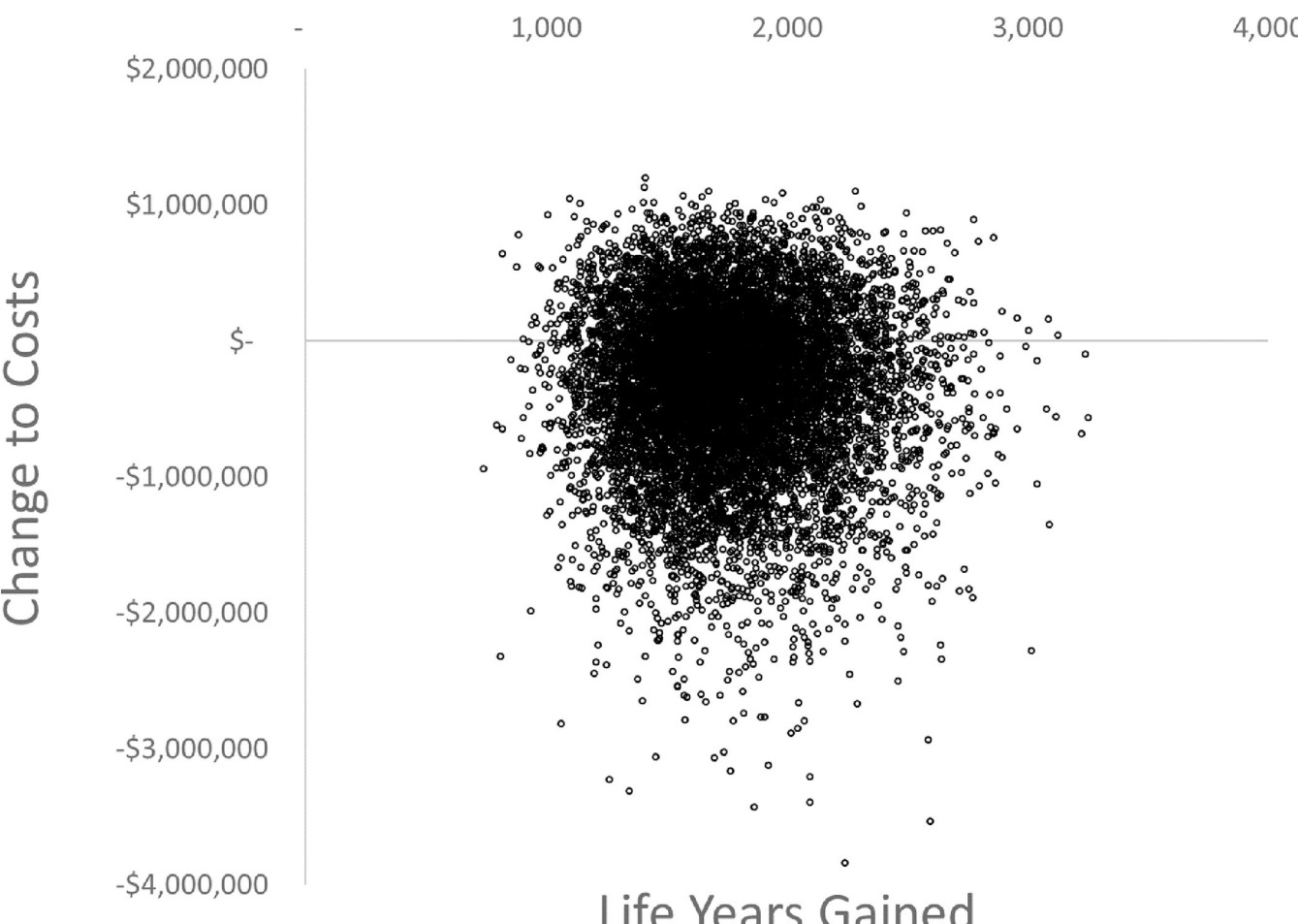

**Fig 1. Joint distribution of the expected change to total costs and health benefits from a decision to adopt temporary isolation rooms, per 100,000 occupied bed days.**

The lowest possible value for the effectiveness of the intervention is that 1% of cases of HAI are prevented; the impact of this assumption on the results are shown in Table 4. On average there will be 16 fewer cases of HAI per 100,000 occupied bed days, 54 bed-days are released for other uses, 4.57 lives will be saved and 59 life years gained. The mean change to total costs is $968,967 for a return of 59 years of life. The incremental cost per life year saved is $16,519.

**Table 4. Changes to outcomes with 1% effectiveness used.**

| Mean (sd) | Cases HAI | bed days | Money value of bed days (SGD) | number of deaths | life years |
|---|---|---|---|---|---|
| ALL HAI | 16 (1) | 54 (7) | $44,518 ($16,022) | 4.57 (0.43) | 59 (6) |
| Sepsis | 5 (0) | 4 (2) | $3,488 ($2,117) | 1.26 (0.22) | 16 (3) |
| Pneumonia | 4 (0) | 11 (2) | $9,329 ($3,728) | 1.53 (0.24) | 20 (3) |
| Surgical | 4 (0) | 14 (3) | $11,297 ($4,433) | 0.75 (0.17) | 10 (2) |
| CLABSI | 1 (0) | 4 (2) | $3,293 ($1,781) | 0.34 (0.12) | 4 (1) |
| Intraabdominal | 1 (0) | 18 (5) | $14,976 ($6,443) | 0.34 (0.11) | 4 (1) |
| Others | 1 (1) | 3 (2) | $2,134 ($1,798) | 0.34 (0.20) | 4 (3) |

CLABSI = central line associated bloodstream infection

There is a zero probability that adoption will be cost saving but a 100% probability that adoption will be cost effective. When mortality benefits are additionally halved the ICER increases to $33,190 per life year gained and the probability that adoption is cost effective remains at 100%. For these scenarios the conclusion is that adoption is a cost-effective decision.

## Discussion

The findings reveal the adoption of a temporary 'pop-up' isolation room only needs to reduce the cases of healthcare acquired infection by 1% to be a cost-effective decision in Singapore public hospitals. It is likely that the real world effectiveness will exceed this, and so the economic benefits will likely be larger. If adoption achieves a 30% reduction in cases, the expectation is that health services costs would reduce by approximately $330,000 per 100,000 bed days, and there would be many lives saved and substantial health benefits. Who actually enjoys the benefit from the cost savings will depend on the funding model of the hospital and the country, but it is likely that hospitals, government funders and patients themselves would benefit.

A strength of this study is that we included a full economic evaluation for a potentially important technology, which considered both the costs to hospitals and health benefits to patients, and quantified and presented the value of an adoption decision with transparency. This contrasts the majority of the infection prevention and control economic evaluations published in literature, which are often partial evaluations of only hospitalisation costs [26]. While we only compared the decision to adopt to 'existing practices', our analyses can be expanded to additionally consider other novel infection prevention and measures.

This study is based on assumptions applied to a model which has limitations compared to a prospective, pragmatic randomised trial. Yet this design would be impossible to blind and complex and slow to implement. Furthermore the time taken would possibly realise opportunity costs in lost savings and lost health gains [27]. In this study we did not consider other factors which could affect implementation. It is possible that there could be a net loss of total beds in a shared cubicle where the typical distance between beds is 1.5 metres. User acceptability will also impact the success of a strategy featuring temporary isolation rooms which must be aesthetically appropriate, comfortable, functional and not associated with stigma. The rooms need to work within nursing workflows, allied health, medical and portering requirements. The advantages of temporary 'pop-up' isolation room as compared to making permanent building reconfigurations are most likely related to costs and speed of deployment.

The quality of the data used for the model parameters is good, with the data gathered for the first national prevalence survey [14] utilised for this analysis. The excess length of stay parameters were estimated using a state-based model that appropriately includes the timing of key events of HAI, death and discharge from hospital [20]. A recent review found analyses that use time fixed methods for the estimation of these outcomes generate biased, inflated, outcomes [28]. The estimates of excess mortality due to infection are naïve as they were not adjusted for other known factors associated with increased mortality. For instance, patients who die with an infection are likely older, with more severe disease and more comorbidities compared to those without HAIs. To address this, we conducted a scenario analysis that halved the probability of death, which in effect halved the health benefits estimated by the model, and found that our conclusions regarding cost-effectiveness were maintained.

As our analyses were robust to uncertainty arising from model parameters and to plausible scenarios, we conclude that our study provides some evidence that the that the adoption of a temporary 'pop-up' isolation room is likely to be cost-effective to Singapore public acute care hospitals, and may potentially result in reduction of healthcare costs. It is plausible that

adoption of this technology is a good decision for other countries in the region, where infection prevention infrastructure is less developed and unlikely to advance in the short or medium term.

## Supporting information

**S1 Checklist. CHEERS 2022 checklist.**
(DOCX)

**S1 Appendix.**
(DOCX)

## Acknowledgments

We are Grateful to Dr Kalis Marimuthu and the steering committee of the HAI PPS surveillance. We are grateful to Kyaw Zaw Linn, Surveillance Coordinator at National Center for Infectious Diseases for organising summary data.

## Author Contributions

**Conceptualization:** Nicholas Graves, Brett Mitchell, Martin Kiernan.

**Data curation:** Nicholas Graves, Yiying Cai, Brett Mitchell, Dale Fisher, Martin Kiernan.

**Formal analysis:** Nicholas Graves, Yiying Cai, Brett Mitchell, Dale Fisher.

**Investigation:** Nicholas Graves, Brett Mitchell, Martin Kiernan.

**Methodology:** Nicholas Graves, Yiying Cai, Brett Mitchell.

**Software:** Nicholas Graves.

**Writing – original draft:** Nicholas Graves, Yiying Cai, Brett Mitchell, Dale Fisher, Martin Kiernan.

**Writing – review & editing:** Nicholas Graves, Yiying Cai, Brett Mitchell, Dale Fisher, Martin Kiernan.

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
