## [Decision Letter · Decision Letter 0]

24 May 2022

PONE-D-22-11005Cost Effectiveness of Temporary Isolation Rooms in Acute Care Settings in AsiaPLOS ONE

Dear Dr. Graves,

Thank you for submitting your manuscript to PLOS ONE. After careful consideration, we feel that it has merit but does not fully meet PLOS ONE’s publication criteria as it currently stands. Therefore, we invite you to submit a revised version of the manuscript that addresses the points raised during the review process.

We look forward to receiving your revised manuscript.

Kind regards,

Monika Pogorzelska-Maziarz

Academic Editor

PLOS ONE

Journal Requirements:

 [No grant funding was used for this project. NG was paid consulting fees by GAMA Healthcare to develop a model and prepare a first draft of the manuscript.] 

[Martin Kiernan is clinical director for GAMA Healthcare.] 

5. PLOS requires an ORCID iD for the corresponding author in Editorial Manager on papers submitted after December 6th, 2016. Please ensure that you have an ORCID iD and that it is validated in Editorial Manager. To do this, go to ‘Update my Information’ (in the upper left-hand corner of the main menu), and click on the Fetch/Validate link next to the ORCID field. This will take you to the ORCID site and allow you to create a new iD or authenticate a pre-existing iD in Editorial Manager. Please see the following video for instructions on linking an ORCID iD to your Editorial Manager account: https://www.youtube.com/watch?v=_xcclfuvtxQ.

Reviewers' comments:

Reviewer's Responses to Questions

**Comments to the Author**

1. Is the manuscript technically sound, and do the data support the conclusions?

Reviewer #1: Yes

Reviewer #2: Partly

Reviewer #3: Yes

2. Has the statistical analysis been performed appropriately and rigorously? 

Reviewer #1: Yes

Reviewer #2: No

Reviewer #3: Yes

3. Have the authors made all data underlying the findings in their manuscript fully available?

Reviewer #1: Yes

Reviewer #2: Yes

Reviewer #3: Yes

4. Is the manuscript presented in an intelligible fashion and written in standard English?

Reviewer #1: Yes

Reviewer #2: Yes

Reviewer #3: Yes

5. Review Comments to the Author

Reviewer #1: The paper is well written. The data analysis for cost effectiveness is appropriate. However, the analyses was mainly based on the hypothesis to reduce HAIs at 30% from the previous studies and the cost was based on the retrospective data, which leads to the lack of reality.

Reviewer #2: The paper analyzes the cost-effectiveness of a decision to adopt temporary isolation rooms in acute care settings in Asia. The paper is technically sound using publicly available data and seeking to offer an alternative low-cost strategy for managing healthcare-associated infections, but not without some infractions. To improve the manuscript for publication, the authors must resolve the following:

1. Title and Abstract

The title suggests a broader spatial context, but the analysis is limited to one country Singapore and not Asia in general, as portrayed in the title. Compare sentence lines 3 and 4 of the abstract to the title.

The method in the abstract should be re-written for clarity. For example, we performed a model-based cost-effectiveness analysis to evaluate ………………. Outcome measures include……………….. OR We measured outcome XXX by doing ABCD and outcome XXX2 by doing MNOP. Etc.

Again, the conclusion that the intervention is cost-effective for Singapore acute care hospitals is misleading because Figure 1 shows that the cost per life-year gain with the intervention may be positive or negative. Thus, about a 50% chance of cost-saving, meaning the intervention is not a dominant strategy.

2. Methodology

In Table 1, the HAI cases reported were inconsistent with the figures documented in citation No.14. Moreso same reference did not categorically state the number of survival and deaths attributable to HAIs as referenced in Table 1 of the manuscript. Since this paper is a model-based paper, one will expect that the cited reference reports the cited baseline figures, and where they are not, provide an explanation for clarity. For instance, the supplementary file attached to the published citation clarified how the paper authors measured some variables but not for mortality attributable to HAI. (Check Table 2 of citation No.14 to resolve the inconsistencies or provide an explanation for clarity).

The authors decided to limit the study horizon to 1 year and attributed the same for not using a preference utility weight to show QALY. In my opinion, it is the author’s discretion not to expand the study horizon, but to attribute the same for not doing additional analysis is misplaced.

3. Scenario analysis.

“The attributable mortality is unadjusted for other factors that can affect mortality”. The authors believed by halving the estimated HAI-attributable mortality, they have ensured robustness, how? They should provide a reference to justify that statement. In the first place, I did not find the mortality figures in the referenced citation used in Table 1 and the explanation of how HAI mortality was arrived at should be provided.

No mention of heterogeneity characterization. It is obvious that the study subjects vary by type of HAIs with different baseline characteristics or observed variabilities which may affect the study conclusion. An aggregated probability risk of HAIs and mortality with 95% uncertainty intervals should be explored in a deterministic sensitivity analysis (DSA) and the result presented in a Tornado graph to offer more explanation.

At baseline, I assume the mean cost per bed day of SDG 823 is an aggregated mean for all the reported HAI cases (Table 2). How about exploring the HAI case-specific mean cost in the scenario analysis to show which contributes more to the cost of HAIs and consequently the cost savings per LYG? I think the same may apply to the length of stay. Obviously, the mean cost of managing surgical site infection is different from the mean cost of managing either CLABSI, Sepsis, etc.

An alternative will be to collapse Table 2 and present only the aggregated mean parameter values, i.e., Mean cost of HAIs, LOS, risk of HAI, etc. with their 95%CI explored in a scenario analysis.

4. Results and discussion

It is the norm that all parameter values, especially those contributing significantly to the study outcome should be captured in the result and discussion section under both scenarios.

Limiting the cost to only the provider perspective results in underestimation of the potential cost associated with HAIs and this must be discussed. In paragraph two of the discussion, the authors align the strength of their paper to doing a full economic evaluation. The statement is not true because the study perspective is limited, and the methodology is not rigorous enough.

5. General comment

The use of study reporting guidelines is highly recommended for quality reporting assurance. In this case, following the CHEERS checklist is a better way to improve the methodology, results, and discussion sections, and it must be stated clearly in the methods the reporting guideline used in this study.

Reference list No 17 is incomplete and others must be checked for proper formatting.

The title of Table 1 should be checked for type error “with and with HAI”

Also, resolve the formatting of the Abstract and figure in line with the Journal requirements.

Reviewer #3: This is a good piece of work that is more relevant to the current resource management of many countries. Most of the countries prepared temporary isolation rooms in their hospitals to isolate COVID 19 patients. Therefore this is a good idea to reduce cost of managing hospital acquired infections using these temporary isolation rooms. At the same time this cost effectiveness study is a technically sound study . Therefore this is a valuable timely performed study.

6. PLOS authors have the option to publish the peer review history of their article (what does this mean?). If published, this will include your full peer review and any attached files.

Reviewer #1: No

Reviewer #2: No

Reviewer #3: No

---

## [Author Response · Author response to Decision Letter 0]

23 Jun 2022

Thanks for the opportunity to respond to the reviewers comments

++++++++

Thank you for stating the following financial disclosure: 

“No grant funding was used for this project. NG was paid consulting fees by GAMA Healthcare to develop a model and prepare a first draft of the manuscript”

Response 1. Here is an amended statement. No grant funding was used for this project. NG was paid consulting fees by GAMA Healthcare to develop a model and prepare a first draft of the manuscript. Martin Kiernan supported many aspects of the study and write up, but he did this without bias and focused on the best interpretation of the data. No other individual from GAMA healthcare influenced the study methods, findings or interpretation. 

Thank you for stating the following in the Competing Interests section: 

Martin Kiernan is clinical director for GAMA Healthcare.

Response 2. Here is an amended competing interest statement. Martin Kiernan is clinical director for GAMA Healthcare. This does not alter our adherence to PLOS ONE policies on sharing data and materials.

In your Data Availability statement, you have not specified where the minimal data set underlying the results described in your manuscript can be found. 

Response 3. Here is an amended statement. “All the data used in the study are reported in the manuscript in summary form. The raw HAI PPS surveillance data are not available for public sharing due to privacy considerations.”

5. PLOS requires an ORCID iD for the corresponding author in Editorial Manager on papers submitted after December 6th, 2016.

Response 4. My orcid ID is assigned to n.graves@qut.edu.au (an old email address). I cannot reassign it to n.graves@duke-nus.edu.sg. Can you help?

Reviewer #1: The paper is well written. The data analysis for cost effectiveness is appropriate. However, the analyses was mainly based on the hypothesis to reduce HAIs at 30% from the previous studies and the cost was based on the retrospective data, which leads to the lack of reality.

Response 5. In addition to assuming 30% of cases of healthcare associated infection will be avoided, we analysed effectiveness by reducing the estimate in the model until the decision to adopt is not supported against the criterion of cost effectiveness. We found the adoption of a temporary ‘pop-up’ isolation room only needs to reduce the cases of healthcare acquired infection by 1% to be a cost-effective decision in Singapore public hospitals. 

Reviewer #2: The paper analyzes the cost-effectiveness of a decision to adopt temporary isolation rooms in acute care settings in Asia. The paper is technically sound using publicly available data and seeking to offer an alternative low-cost strategy for managing healthcare-associated infections, but not without some infractions. To improve the manuscript for publication, the authors must resolve the following:

1. Title and Abstract

The title suggests a broader spatial context, but the analysis is limited to one country Singapore and not Asia in general, as portrayed in the title. Compare sentence lines 3 and 4 of the abstract to the title.

Response 6. The title has been changed to this. “Cost Effectiveness of Temporary Isolation Rooms in Acute Care Settings in Singapore” 

The method in the abstract should be re-written for clarity. For example, we performed a model-based cost-effectiveness analysis to evaluate ………………. Outcome measures include……………….. OR We measured outcome XXX by doing ABCD and outcome XXX2 by doing MNOP. Etc.

Response 7. The methods section in the abstract has been changed. “We performed a model-based cost-effectiveness analysis to evaluate the impact of a decision to adopt temporary isolation rooms for infection prevention. We estimated changes to the costs from implementation, the number of cases of healthcare associated infection, acute care bed days used, they money value of bed days, the number of deaths, and the expected change to life years. We report the probability that adoption was cost-effective by the cost by life year gained, against a relevant threshold.”

Again, the conclusion that the intervention is cost-effective for Singapore acute care hospitals is misleading because Figure 1 shows that the cost per life-year gain with the intervention may be positive or negative. Thus, about a 50% chance of cost-saving, meaning the intervention is not a dominant strategy.

Response 8. Figure 1 reveals a 100% probability that adoption will be cost effectiveness against a threshold of $SGD80,000 per life year gained. It also reveals a 67% probability that adoption will be cost savings and health increasing. The probability that non-adopting is cost effective is zero.

2. Methodology

In Table 1, the HAI cases reported were inconsistent with the figures documented in citation No.14. More so same reference did not categorically state the number of survival and deaths attributable to HAIs as referenced in Table 1 of the manuscript. Since this paper is a model-based paper, one will expect that the cited reference reports the cited baseline figures, and where they are not, provide an explanation for clarity. 

Response 9. The information in Table 1 are reported for the first time. Although collected for the purposes of the prevalence survey - cite [14] - they were not reported in that paper.

For instance, the supplementary file attached to the published citation clarified how the paper authors measured some variables but not for mortality attributable to HAI. (Check Table 2 of citation No.14 to resolve the inconsistencies or provide an explanation for clarity).

Response 10. The appendix in this manuscript is an example of the R script and output used to estimate attributable mortality. It simply shows the method used to estimate attributable mortality. This is the first time this information has been reported, it was not included in cite [14]

The authors decided to limit the study horizon to 1 year and attributed the same for not using a preference utility weight to show QALY. In my opinion, it is the author’s discretion not to expand the study horizon, but to attribute the same for not doing additional analysis is misplaced.

Response 11. As HAI is a relative short duration event (10 to 14 days) we feel 12 months is adequate to capture the consequences. Weighting outcomes by preference based utility scores would not change the conclusions, because the duration of the illness is so short. QALYs and utility weights are more informative for chronic diseases that take years to resolve. 

3. Scenario analysis.

“The attributable mortality is unadjusted for other factors that can affect mortality”. The authors believed by halving the estimated HAI-attributable mortality, they have ensured robustness, how? They should provide a reference to justify that statement. 

Response 12. All the health benefits arise from mortality avoided from fewer cases of HAI. We suggest the mortality benefits might be overstated somewhat because the attributable mortality is unadjusted for other factors that can affect mortality. We are unable to obtain adjusted estimates, so we took the pragmatic decision to reduce mortality benefits dramatically - by 50% - and test whether the conclusions still supported adoption; and they do. We are fairly certain that the real mortality benefits lie between the baseline estimates and the 50% estimate. Thus we are comfortable our results are robust. 

In the first place, I did not find the mortality figures in the referenced citation used in Table 1 and the explanation of how HAI mortality was arrived at should be provided.

Response 13. See Response 9.

No mention of heterogeneity characterization. It is obvious that the study subjects vary by type of HAIs with different baseline characteristics or observed variabilities which may affect the study conclusion. An aggregated probability risk of HAIs and mortality with 95% uncertainty intervals should be explored in a deterministic sensitivity analysis (DSA) and the result presented in a Tornado graph to offer more explanation.

Response 14. We have modelled patients based on the type of HAI they acquired and the model accounts for this by applying different costs, excess stays and risks of mortality. We prefer to report a probabilistic sensitivity analyses that captures all the parameter uncertainties simultaneously; this approach is more useful and more informative for decision making that using fixed value sensitivity analysis and the associated tornado graphs. Fig 1 provides a lot more information that we might glean from a tornado graph. Probabilistic sensitivity analyses is recommended by the CHEERS checklist. 

At baseline, I assume the mean cost per bed day of SDG 823 is an aggregated mean for all the reported HAI cases (Table 2). How about exploring the HAI case-specific mean cost in the scenario analysis to show which contributes more to the cost of HAIs and consequently the cost savings per LYG? I think the same may apply to the length of stay. Obviously, the mean cost of managing surgical site infection is different from the mean cost of managing either CLABSI, Sepsis, etc.

Response 15. The cost per bed day of 823 is the mean costs or providing an acute bed day in the Singapore healthcare system. The costs of HAI are shown by multiplying the Excess LOS for each HAI - see Table 2 -with this estimate.

An alternative will be to collapse Table 2 and present only the aggregated mean parameter values, i.e., Mean cost of HAIs, LOS, risk of HAI, etc. with their 95%CI explored in a scenario analysis.

Response 16. I believe Table 2 is appropriate for this study. It shows the values used and distributions for all the model parameters. 

4. Results and discussion

It is the norm that all parameter values, especially those contributing significantly to the study outcome should be captured in the result and discussion section under both scenarios.

Response 17. All parameter values used to generate the results are reported in Table 2. 

Limiting the cost to only the provider perspective results in underestimation of the potential cost associated with HAIs and this must be discussed. In paragraph two of the discussion, the authors align the strength of their paper to doing a full economic evaluation. The statement is not true because the study perspective is limited, and the methodology is not rigorous enough.

Response 18. Using a broader perspective will add weight to the already strong conclusions. We see limited value to adding in costs for primary care services or even production losses. It would make the paper bulkier and would only re-enforce the cost-effectiveness result by making the cost savings greater. We believe the methods we report are appropriate and rigorous for the stated research question. 

5. General comment

The use of study reporting guidelines is highly recommended for quality reporting assurance. In this case, following the CHEERS checklist is a better way to improve the methodology, results, and discussion sections, and it must be stated clearly in the methods the reporting guideline used in this study.

Response 19. A CHEERS checklist has been completed. This text has been added to the methods. “A CHEERS checklist has been completed and included as an appendix.”

Reference list No 17 is incomplete and others must be checked for proper formatting.

Response 20. References have been checked and updated

The title of Table 1 should be checked for type error “with and with HAI”

Response 21. This has been fixed.

Also, resolve the formatting of the Abstract and figure in line with the Journal requirements.

Response 22. The abstract meets journal requirements 

Reviewer #3: This is a good piece of work that is more relevant to the current resource management of many countries. Most of the countries prepared temporary isolation rooms in their hospitals to isolate COVID 19 patients. Therefore this is a good idea to reduce cost of managing hospital acquired infections using these temporary isolation rooms. At the same time this cost effectiveness study is a technically sound study . Therefore this is a valuable timely performed study

---

## [Editor Report · Decision Letter 1]

7 Jul 2022

Cost Effectiveness of Temporary Isolation Rooms in Acute Care Settings in Singapore

PONE-D-22-11005R1

Dear Dr. Graves,

We’re pleased to inform you that your manuscript has been judged scientifically suitable for publication and will be formally accepted for publication once it meets all outstanding technical requirements.

Kind regards,

Monika Pogorzelska-Maziarz

Academic Editor

PLOS ONE

---

## [Editor Report · Acceptance letter]

13 Jul 2022

PONE-D-22-11005R1 

Cost Effectiveness of Temporary Isolation Rooms in Acute Care Settings in Singapore 

Dear Dr. Graves:

I'm pleased to inform you that your manuscript has been deemed suitable for publication in PLOS ONE. Congratulations! Your manuscript is now with our production department. 

Kind regards, 

on behalf of

Dr. Monika Pogorzelska-Maziarz 

Academic Editor

PLOS ONE